

# Validation of the near-wake of a scaled X-Rotor vertical-axis wind turbine predicted by a free-wake vortex model

Adhyanth Giri Ajay, David Bensason, and Delphine De Tavernier

Wind Energy section, Flow Physics and Technology department, Faculty of Aerospace Engineering, Delft University of Technology, Kluyverweg 1, 2629HS Delft, The Netherlands.

**Correspondence:** Adhyanth Giri Ajay (a.giriajay@tudelft.nl)

**Abstract.** Vertical-axis wind turbines (VAWTs) are gaining research attention in offshore energy due to their ability to operate in omnidirectional wind, simpler design characteristics, and potential for faster wake recovery. As part of this interest, a novel X-shaped VAWT (X-Rotor) has been proposed to minimise the levelised cost of energy. While existing studies on the X-Rotor rely on numerical tools to analyse rotor performance, experimental validation remains limited, making it essential to assess the accuracy of these models in predicting the flowfield around the rotor. This study compares a free-wake vortex model (CACTUS) against stereoscopic particle image velocimetry (PIV) results for a scaled X-Rotor. Both qualitative and quantitative comparisons are performed, examining flowfield features with and without blade pitch offsets. Additionally, the study provides insights into the three-dimensional aerodynamics introduced into the wake by the turbine's coned blades. Results indicate that CACTUS effectively replicates the flowfield within the rotor volume and the very near wake when no pitch offsets are applied. However, with pitch offsets, significant deviations from experimental data are observed, suggesting the need for careful model tuning for full-scale X-Rotor analysis. Furthermore, the introduction of coned blades enhances three-dimensional effects, generating notable upwash and downwash in the wake. These findings highlight the importance of using 3D aerodynamic tools over 2D approaches in future X-Rotor analyses to accurately capture vertical flow components.

## 1   Introduction

Over the last decade and a half, vertical-axis wind turbines (VAWTs) have been considered an attractive alternative to horizontal-axis wind turbines (HAWTs). This is due to their omnidirectional operation, simpler mechanical systems, lower centre of mass, and potential for better performance in urban and offshore environments (Dabiri, 2011; Ishugah et al., 2014; Tjiu et al., 2015; Su et al., 2020; Lee et al., 2022). According to Lee and Zhao (2021), the rate of wind energy deployment needs to increase threefold by 2030 to meet our climate goals. In this context, several novel VAWT designs are currently being developed and researched. Among these, an X-shaped VAWT configuration, named X-Rotor, has been designed to lower the LCoE for offshore applications (Leithead et al., 2019).

The X-Rotor geometry (Figure 1) consists of two main components: an X-shaped VAWT (primary rotor) and two tip-mounted HAWTs (secondary rotors). The primary rotor has a distinctive 'X' shape, with a set of coned upper and lower blades connected by a cross-beam that serves as a strut. The secondary rotors are attached at the lower blade tips and experience a





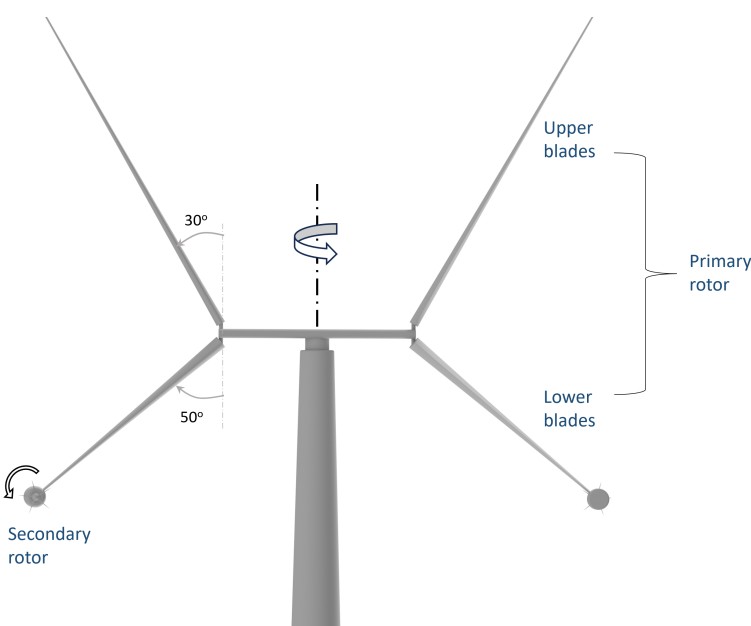

**Figure 1.** A geometric render of the X-Rotor with the different components.

significantly increased inflow speed due to the relative velocity at the primary rotor blade tips. The thrust of the secondary rotors determines the rotational speed of the overall turbine, and all electrical power is extracted from them. These rotors are connected to high-speed direct-drive generators instead of using gearboxes, which substantially reduces the turbine's capital costs. Additionally, the low altitude and reduced mass of the generators eliminate the need for jack-up vessels for maintenance, potentially lowering operations and maintenance expenses (Flannigan et al., 2022). The upper blades of the X-Rotor are pitch-
controlled and designed to shed aerodynamic power in above-rated conditions (Recalde-Camacho et al., 2024). The lower blades are not pitch-controlled, as any adjustment to them would disrupt the operation of the secondary rotors. A recent study on the operational expenditure of the X-Rotor concept by Flannigan et al. (2022) demonstrated significant savings in the operational cost of energy compared to a HAWT. Similarly, a feasibility study by (Leithead et al., 2019) showed up to 26% overall cost savings compared to HAWTs.

Existing aerodynamic studies on the X-Rotor geometry are quite limited. Morgan and Leithead (2022) provided an initial characterisation of the X-Rotor using a double multiple streamtube (DMS) method. Later, Morgan et al. (2025) demonstrated the power gains achieved by coned blades compared to non-coned blades for a given blade span. In our earlier work, we systematically compared aerodynamic models—ranging from low-fidelity BEM models to high-fidelity blade-resolved URANS CFD models—on the power, thrust, and load characteristics of a full-scale X-Rotor at various operational blade pitch offsets
(Giri Ajay et al., 2024). We concluded that low-fidelity BEM models, such as DMS and 2DAC, are unsuitable for modelling the X-Rotor across its full operating range due to vertical induction from coned blades and stronger tip vortices under pitch-offset conditions. Furthermore, by comparing with high-fidelity models, we showed that free-wake vortex models are a promising





alternative, offering a relatively cost-efficient approach while capturing the effects of vertical induction. However, in all these existing studies on the X-Rotor, the models were only compared against higher-fidelity simulations in limited operational cases

and could not be validated with experimental data due to the full-scale rotor size.

Free-wake vortex models have previously been experimentally validated in the near wake for HAWTs (Sant et al., 2005; Gupta and Leishman, 2006; Van Den Broek et al., 2023). Similar validations have been conducted for H-type VAWTs (Ferreira et al., 2010; Meng et al., 2014; Tescione et al., 2016), demonstrating that free-wake vortex models are highly effective in accurately capturing the VAWT's near wake.

Recently, two experimental campaigns of a 1:100 scaled X-Rotor geometry (Figure 1) were conducted. Bensason et al. (2023) obtained phase-locked PIV data of the induction field of the rotor with no pitch offsets. The dataset consisted of planar slices at different locations inside the rotor volume to examine the influence of the coned blades on the vorticity field. Later, Bensason et al. (2024) captured additional phase-locked data in the near wake of the turbine with pitch offsets, focusing on their impact on the near-wake flow.

This study aims to validate the aerodynamic characteristics of the scaled X-Rotor predicted by CACTUS by comparing numerical simulations with wind tunnel measurements for both non-pitched and pitched cases. Furthermore, it examines the influence of the coned blades on blade loads and the flow field as a function of rotor height. Since wake and vortex structures depend on blade loads, this study strengthens confidence in using free-wake vortex models for unconventional turbines. Additionally, it provides valuable insights into the X-Rotor's aerodynamics, supporting its future development.

## 2 Experimental approach

In this section, we provide a brief description of the rotor model used in the experiments and the experimental setup. Detailed information on setup and measurement techniques can be found in Bensason et al. (2023) and Bensason et al. (2024).

### 2.1 Scaled X-Rotor model

The test geometry is a geometrically scaled model of the full-size primary rotor, reduced by a factor of $\frac{1}{100}$. The top and bottom
blades have a tip diameter of $D = 1.5$ m and cone angles of $30°$ and $50°$, respectively, resulting in upper and lower spans of 1 m and 0.65 m, respectively. Aside from scaling, the primary difference between the full-scale turbine and the scaled model is that the latter consists of four straight NACA0021 airfoils with a constant chord of $c = 0.075$ m, attached to a stiff crossbeam of length 0.5 m with the same profile and chord. The rotor is supported by a tower with a diameter of 0.06 m. The model operates at a constant tip-speed ratio $\lambda = 4.0$, yielding a chord-based Reynolds number of $Re_c = 8.1 \times 10^4$ at the tip.

A visualisation of the tip-trajectory is provided in Figure 2. A constant inflow velocity of $U_\infty = 4$ m/s is assumed, neglecting the influence of induction. Blades (B) 1 and 2 represent the blades at azimuth $\theta = 0°$ and $180°$, corresponding to the rotor's perpendicular position to the flow (maximum frontal area), as labelled in Figure 2. The top (t) and bottom (b) blade pairs follow the same convention. The structures associated with each blade (B1t, B2t, B1b, B2b) are distinguished by colour. As the blades progress over time, the resulting flow structures from each blade convect downstream, eventually overlapping and



interacting with others. Visualisations of each phase are provided here as a reference for analysing the flow fields at the specified streamwise locations within the volumes.

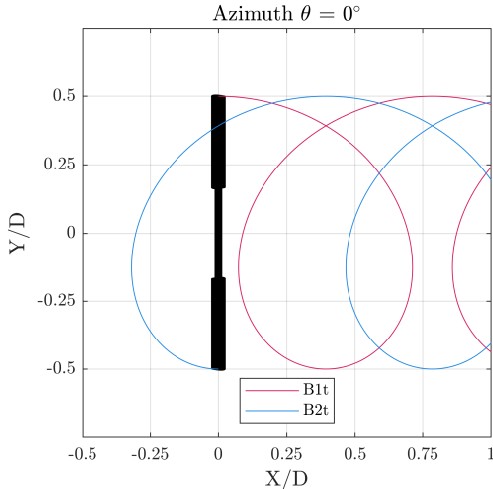

**Figure 2.** Top-view schematic of tip-trajectory of the top blades at azimuth $\theta = 0°$. The tip-trajectory are mapped to each blade, in this instance labelled B1t and B2t, for the two blades in the top-half (t). Corresponding blades in the bottom half would be referred to as B1b and B2b respectively. No wake expansion or induction is assumed.

## 2.2 Experimental set-up

The experiments are conducted in the Open Jet Facility (OJF) at TU Delft Aerospace Engineering, as illustrated in Figure 3. A controlled streamwise velocity of $U_\infty = 4$ m/s is maintained throughout the experiment. The resulting inflow has reported

turbulence intensities of 0.5% within the testing region Lignarolo et al. (2014). The measurement system captures phase-locked flow field measurements at various locations within the X-Rotor induction field. Normalised cross-stream locations $x/D = -0.435, -0.265, -0.065, 0, 0.065, 0.265, 0.435$ were measured at azimuths $\theta = \{0°, 45°, 90°, 135°\}$. At each cross-stream location, multiple planes are recorded along the y- and z-axis using a traversing system and then stitched together. Due to time constraints, not all phase and wake locations were measured with the same level of detail, leading to variations

in the number of planes recorded for each wake-phase pair. Additionally, because of the camera orientation relative to the measurement planes, frequent masking operations are applied to eliminate shadows. Given the placement of the PIV system, the measurements primarily focus on the windward half of the cycle ($Y/D > 0$).





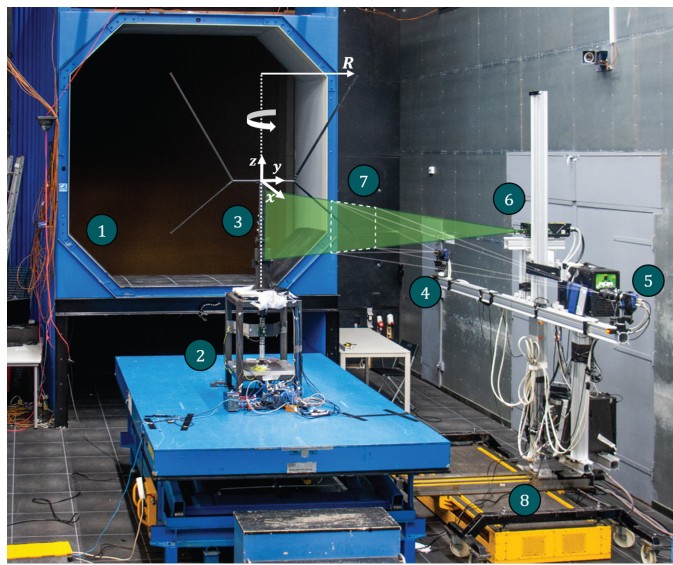

① OJF outlet ② Turbine base ③ X-Rotor model ④ Camera 1
⑤ Camera 2 ⑥ Laser ⑦ Field-of-view ⑧ Traversing system

**Figure 3.** Experimental setup in the OJF from adapted Bensason et al. (2023). The origin of the coordinate system is placed at the center of the crossbeam. The axis and direction of rotation are marked (anti-clockwise) with a curved arrow, and the tip radius $R$ is marked. A visualization of the measurement plane is provided (green cone). A similar setup for the second experimental campaign is presented in Bensason et al. (2024).

### 2.3 Uncertainty of the flowfield measurements

The diffraction-limited minimum image diameter is of practical significance for optical measurements such as PIV. Following Equation (1), the smallest particle image that can be obtained using the given imaging configuration is defined by $d_{\mathrm{diff}}$ Adrian and Westerweel (2011).

In this case, the magnification factor is $M = 0.01$ (sCMOS camera catalogue, 6.5 $\mu$m). The lens used has a focal length $f$ of 106 mm, resulting in an object imaging distance ($d_o$) of 101 mm. The f-number $f_{\#}$ for this experiment is set to 8, and the sCMOS cameras have a wavelength of $\lambda = 523$ nm (catalogue). Consequently, $d_{\mathrm{diff}}$ is 10.1 $\mu$m. Given a particle pixel diameter $d_p$ of approximately 3 $\mu$m, the ratio $d_{\mathrm{diff}}/d_p$ is 3.4.

$$d_{\mathrm{diff}} = 2.44 \times f_{\#} \times (M + 1) \times \lambda \tag{1}$$

We can also examine the standard uncertainty in the mean flow field. Using the approach of Sciacchitano and Wieneke (2016), the standard uncertainty is calculated with Equation (2). For each measurement, we average over $N = 120$ samples. The standard deviation of the velocity, $\sigma_{\mathrm{U}}$, consists of all three velocity components and varies across the plane, both temporally and spatially. In general, regions with high vorticity (such as tip vortices and shed vorticity) exhibit a velocity standard deviation



of $\sigma_U = 0.24$ m/s, with the highest component occurring out of the plane, while the other two components are 0.19 m/s and 0.15 m/s. In wake regions where no vortical structures are present, the velocity standard deviation is lower, approximately $\sigma_U = 0.05$ m/s, with all components being similar in magnitude. Using the highest value for calculation, the standard uncertainty of the mean velocity is $U_U = 0.02$ m/s, which corresponds to 0.5% of the free-stream velocity.

$$U_U = \frac{\sigma_U}{\sqrt{N}} \tag{2}$$


A similar uncertainty analysis is conducted for the cases with blade pitch offset and is presented in detail in Bensason et al. (2024).

## 3 Numerical setup

### 3.1 CACTUS free-wake vortex model (CACTUS)

The Code for Axial and Cross-flow TUrbine Simulation (CACTUS) is an open-source vortex model tool for wind turbine simulations, developed by Murray and Barone (2011). In this approach, the blades are represented as a lifting line, with a vortex lattice system formed by the surrounding bound, shed, and trailing circulations (Figure 4).

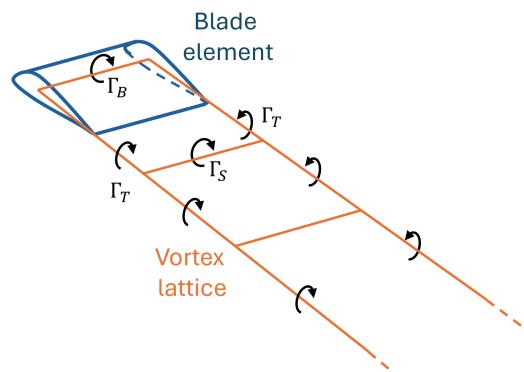

**Figure 4.** The vortex lattice system of a single blade element along with the bound ($\Gamma_B$), shed ($\Gamma_S$), and the trailing ($\Gamma_T$) circulations.

The flow field is constructed using this vortex lattice, where the velocity is the arithmetic sum of the freestream velocity and the velocity induced by the vortices. This is determined using the Biot-Savart law (Katz and Plotkin, 2009), which expresses

the velocity field as a function of the circulation of any line segment within the vortex lattice, as follows:





$$\boldsymbol{u_a} = \frac{\Gamma}{4\pi} \frac{\boldsymbol{r_1} \times \boldsymbol{r_2}}{|\boldsymbol{r_1} \times \boldsymbol{r_2}|^2} (\boldsymbol{r_1} - \boldsymbol{r_2}) \cdot \left( \frac{\boldsymbol{r_1}}{|\boldsymbol{r_1}|} - \frac{\boldsymbol{r_2}}{|\boldsymbol{r_2}|} \right) \tag{3}$$

Here, $\boldsymbol{u_a}$ represents the velocity vector at point $\boldsymbol{a}$, while $\boldsymbol{r_1}$ and $\boldsymbol{r_2}$ are the displacement vectors from the ends of the vortex line segment to point $\boldsymbol{a}$. The circulation of the vortex line segment is denoted by $\Gamma$.

In this study, we used CACTUS to model the scaled X-Rotor, representing each of the upper and lower blades as lifting lines. Additionally, we implement the free-wake vortex algorithm in CACTUS, which calculates the wake convection velocity at each time step. To model the circulation distribution, the NACA0021 blade airfoil profile is characterised using an airfoil polar dataset, which is further detailed in Section 3.2.

### 3.2 NACA0021 airfoil dataset

In our study, the experiments operate at a chord-based Reynolds number of $Re = 80 \times 10^3$. Melani et al. (2019) collected experimental and numerical polars for the NACA0021 airfoil across different Reynolds number ranges: low $(40 - 80 \times 10^3)$, medium $(80 - 700 \times 10^3)$, and high $(\geq 700 \times 10^3)$. In the low ranges, there is a significant deviation among the polar profiles, due to measurement uncertainties. In contrast, in the medium range, the polars exhibit much better agreement among each other and show only a marginal increase in peak lift coefficient $C_l$ compared to the low range. Given this, we opted to run the simulations at the medium range of $Re = 150 \times 10^3$ instead of using the operating Reynolds number. This choice introduces only minor changes to the polar slope and static stall angle but significantly improves the reliability of the aerodynamic loads and overall simulation results.

In this study, we generated the airfoil polars using XFOIL (Drela, 1989) at the previously mentioned Reynolds number of $Re = 150 \times 10^3$ for the X-Rotor, with an Ncrit of 8. This set of polars is the same as that used in Bensason et al. (2024), where blade loads were calculated using CACTUS to validate the actuator cylinder theory on the influence of loads on wake topology (Ferreira, 2009). The static stall angle in this case is $\alpha_{ss} = 9.1°$.

### 3.3 Dynamic stall and flow curvature models

Dynamic stall models are used to evaluate the unsteady effects on the lift, pitching moments, and drag of the blade sections, enabling more accurate predictions from the free-wake vortex model (Masson et al., 1998; Buchner et al., 2018; Le Fouest and Mulleners, 2022). Considering the operational Reynolds number of the scaled X-Rotor, we employ the Leishman-Beddoes (LB) dynamic stall model (Sheng et al., 2006, 2008), which is readily available in CACTUS. The LB model simulates the dynamic stall process by solving a set of first-order differential equations and utilising empirical data to represent delayed flow separation, vortex shedding, and hysteresis in lift and drag.

VAWT airfoils undergo circular motion over their azimuthal cycle, which can be decomposed into translation and pitching motion. This pitching component leads to a varying inflow not experienced by HAWTs, which introduces flow curvature effects that must be compensated for by an additional angle of attack or virtual camber (Migliore et al., 1980; Cardona, 1984;



Rainbird et al., 2015). While this provides a more accurate depiction of airfoil motion in VAWTs, we opt not to include it here, as implementing it with a spanwise distribution that has not been tested could introduce unwarranted errors into our analysis.

### 3.4 Case matrix and simulation procedure

150 To simulate the X-Rotor, the upper and lower blades are discretised into 18 blade sections each, which is the minimum required to achieve blade element independence of power. An additional blade section is included to smooth the transition between the upper and lower blades with fixed pitch offsets. The cross-beam and tower are not modelled, as their contributions are expected to be minimal and predictable, offering little additional insight into the aerodynamics of the X-Rotor. A constant vortex core model is employed, with the vortex core set to 100% of the chord-to-radius ratio. The simulations are run for 10 revolutions to ensure convergence, using a second-order predictor explicit time advancement scheme. A sensitivity analysis was conducted 155 on vortex core size, dynamic stall, and the effects of lift and drag coefficients for the airfoil. To maintain focus on the main findings, this analysis is presented separately in Appendix A. The conclusions drawn from the sensitivity study informed the final setup parameters summarised in Table 1. The solidity $\sigma$ of the X-Rotor is determined using the derived expression: $\sigma = \frac{Nc(L_U \cos 30° + L_L \cos 50°)}{A}$, where $N$ is the number of blades, $L_U$ and $L_L$ are the upper and lower blade spans, $c$ is the chord, and $A$ is the rotor frontal area. The angles correspond to the coned angles of the upper and lower blades.

**Table 1.** Description of the simulation setup for CACTUS

| Parameters | Value |
|---|---|
| Rotor diameter (at the tips) $D$ | 1.5 m |
| Rotor chord $c$ | 0.075 m |
| Upper blade span $L_U$ | 1 m |
| Lower blade span $L_L$ | 0.65 m |
| Frontal Area $A$ | 1.287 m$^2$ |
| Number of blades $N$ | 2 |
| Airfoil profile | NACA0021 |
| Tip-speed ratio $\lambda$ | 4 |
| Inlet velocity $U_\infty$ | 4 m/s |
| Air density $\rho$ | 1.207 kg/m$^3$ |
| Solidity $\sigma$ | 0.15 |
| Blade elements (per blade) | 18 |
| Azimuthal discretisation | 72 |
| Blade pitch $\beta$ | $-10°, 0°, 10°$ |
| Vortex core size (fraction of local chord) | 1 |



## 4 Results

### 4.1 Validation study - non-pitched case

The streamwise velocity $u/U_\infty$ and vorticity $\omega_x D/U_\infty$ contours at different azimuthal positions $\theta = [0°, 45°, 90°, 135°]$ are compared between CACTUS simulations and the experimental outputs in Figure 5 and Figure 6 respectively. The planes are located at $X/D = -0.27, 0.067, 0.27, 0.43$ and showcase the simulations in the upper tile and the experimental results in the lower tile for each phase. The plane $X/D = 0$ was not measured in the experiment as it produced considerable reflection inducing high uncertainty in the PIV processing.

Overall, the velocity predictions within the rotor volume show good agreement with the experimental results, as observed in Figure 8, except for the wake of the tower, which is not modelled in CACTUS. The flowfield trends align well with experimental data, with velocity deficits accurately represented in the regions of blade passage across all azimuths. The induction in the upwind plane at $X/D = 0.27$ is also well predicted. Some minor discrepancies are expected, as CACTUS does not account for blade deflection and deformation caused by centrifugal forces and the high rotational speed of the rotor. This is particularly noticeable at the windward tips ($Y/D > 0$), where the experimental results show velocity deficits extending beyond the projected frontal area of the rotor—an effect not captured in CACTUS. At $X/D = 0.43$, CACTUS slightly under-predicts the velocity deficit across all azimuths. This difference is likely due to the vortex core size being larger than necessary to resolve fine vortical structures. However, attempts to reduce the vortex core size led to stability issues in the X-Rotor model (Appendix A). Consequently, the results presented here represent the most stable configuration achieved with the smallest viable vortex core sizes. A similar conclusion was drawn by Mendoza et al. (2019) in LES validation studies using actuator line methods.

A similar trend is observed in the vortex structures (Figure 9), with the simulation results generally aligning well with the experiments. The elliptical vortical structures seen in the experiments correspond to the shed vortices from the blades, as detailed in Bensason et al. (2023). These vortices take on an elliptical shape due to the coned geometry of the X-Rotor blades. While the shed vortices are captured in the CACTUS model, they appear as faint smears (e.g., in the upper half at $\theta = 0°$ and $X/D = 0.067$) and are significantly weaker in magnitude compared to the experimental results. This discrepancy is once again attributed to the choice of vortex core size; a smaller core size would better resolve the shed vortices as observed in the experiments. The isosurfaces further illustrate the evolution of dominant vortices in the CACTUS simulations. Comparing with Figure 2, the vortices generated at $X/D = -0.27$ at $\theta = 0°$ originate from the passage of B2t and B2b. At $X/D = 0.067$, these vortices intersect the windward tip, while the other dominant vortices are remnants from the previous cycle of B1t and B1b—evident from the isosurfaces, which reveal two sets of trailing vortices convecting downstream. This pattern extends to other planes and phases, highlighting vortex interactions between the blade passages of B1 and B2. At $X/D = 0.43$, the downwind passage of B1t and B1b generates three pairs of tip vortices in both the upper and lower halves of the rotor. This matches the experimental observations, although the absence of leeward planes in the measurements makes it challenging to confirm the presence of these vortex pairs definitively. Additionally, in the experiments, vortices tend to intersect the measurement plane at laterally farther locations than predicted by the simulations (e.g., tip vortices of B2t and B1t at $X/D = 0.067$ and $X/D = 0.43$). This discrepancy arises from blade deflection, which is not accounted for in CACTUS.





**Figure 5.** Normalised streamwise velocity ($u/U_\infty$) contours of the X-Rotor at azimuths $\theta = [0°, 45°, 90°, 135°]$ respectively at downstream locations $X/D$ = -0.27, 0.067, 0.27, 0.43 where $D$ is the rotor diameter. The numerical and experimental results are shown in the upper and lower tile respectively. The black dash lines indicate the projected frontal area of the rotor on the corresponding plane. The x-axis is magnified to enhance visibility.

As the blade tips deflect outward, the vortices are shed from a more laterally displaced position, affecting their subsequent convection paths.





(a) $\theta = 0°$

(b) $\theta = 45°$

(c) $\theta = 90°$

(d) $\theta = 135°$

**Figure 6.** Normalised streamwise vorticity ($\omega_x D/U_\infty$) contours of the X-Rotor at azimuths $\theta = [0°, 45°, 90°, 135°]$ respectively at downstream locations $X/D$ = -0.27, 0.067, 0.27, 0.43 where $D$ is the rotor diameter. The numerical and experimental results are shown in the upper and lower tile respectively. Isosurfaces generated from the CACTUS simulations are overlayed over it to show the evolution of the vortices and represent vortices with non-dimensionalised strength of 6 and above. The black dash lines indicate the projected frontal area of the rotor on the corresponding plane. Very low values of vorticity are hidden and x-axis is magnified to enhance visibility.

Quantitative comparisons between the CACTUS model and the experimental results can be made by examining streamwise velocity profiles at various streamwise locations, both upstream and downstream of the rotor, across different azimuthal phases.





Figure 7 presents these comparisons, showing streamwise velocity slices $u/U_\infty$ as a function of X-Rotor height, evaluated at $Y/D = -0.125$ and $Y/D = 0.25$ for azimuthal positions $\theta = [0°, 45°, 90°, 135°]$.



**Figure 7.** Comparison of the streamwise velocity $u/U_\infty$ slices along the height of the X-Rotor at lateral position $Y/D = -0.125$ (left) and 0.25 (right) at discrete streamwise locations $X/D$ = [-0.27, 0.067, 0.27, 0.43] at phase locked azimuths $\theta = [0°, 45°, 90°, 135°]$. The inset figure in the first tile is presented to indicate the location of the slice with respect to the rotor - the green line is the $Y/D$ slice along the height of the rotor.

200    As expected from the earlier contour comparisons, the CACTUS results align well with the experimental data. In the leeward slice ($Y/D = -0.125$), CACTUS consistently underpredicts the velocity in the lower half of the turbine, except in the most upwind plane. This discrepancy is attributed to the influence of the tower. In the experiments, the counter-clockwise rotation of the tower induces local flow acceleration on the leeward side, which is not accounted for in the CACTUS model. Since this



region is close to the tower, its influence is more pronounced here than on the windward side. Additionally, CACTUS does not
fully capture some of the velocity spikes observed in the experiments, particularly at $\theta = 45°$ and $\theta = 135°$. This discrepancy
arises from two factors: blade deflection in the experiments and the velocity field resolution in CACTUS. The former affects
vortex proximity to the measurement plane—blade deflection in the experiments can bring vortices closer than simulated, as
seen at $\theta = 135°$. The latter relates to the resolution of the volumetric velocity field obtained from CACTUS. Increasing the
resolution to match the PIV planes would better capture these spikes but would also result in excessively large data files due
to CACTUS's volumetric outputs. Another notable difference is the influence of the struts in the PIV data, which is evident
at $\theta = 45°$ and $X/D = 0.067$. In the windward slice ($Y/D = 0.25$), the differences between CACTUS and the experiments
are lesser than the leeward side. The influence of the tower is minimised as there is no observable trend of overprediction
or underprediction from CACTUS in the lower half of the rotor. The upwind plane has the best match while the differences
increase as we move downwind. At $\theta = 135°$, in the planes $X/D = 0.067$ and $0.27$ have the most observable deviations.
This is due to the blade wake interaction occurring at this azimuth which is under-represented in CACTUS compared to the
experiment. Just as the leeward side, the peaks of velocity profiles occurs at different heights due to the deflection of blade not
being modelled in CACTUS. Once again, the ability to resolve the spikes is largely dependent on the resolution of the velocity
field obtained from CACTUS (notably seen at $X/D = 0.43$ for $\theta = 0°$). Overall, the trends of the wake are captured and the
velocity deficits are predicted well - which indicates that CACTUS can be a good tool to represent the velocities inside the
volume of the X-Rotor.

## 4.2   Validation study - pitched case

Similar to the work by Bensason et al. (2024), a positive pitch corresponds to orienting the leading edge towards the rotation
axis, while a negative pitch directs it away. Additionally, in line with the original control strategy of the X-Rotor design (Leit-
head et al., 2019), the lower blades are not pitched along with the upper blades. The experiments measured planes downwind
of the rotor for all three pitch cases, $\beta = -10°, 0°$ (baseline), and $10°$, at an azimuth of $\theta = 0°$, but only for the upper blades.

To facilitate comparison, Figure 8 and Figure 9 present the normalised streamwise velocity $u/U_\infty$ and vorticity $\omega_x D/U_\infty$
respectively, for the three pitch cases in both CACTUS simulations and experimental results.

The baseline pitch case effectively extends the analysis from Figure 5 and the discussion in Section 4.1. While the CACTUS
model continues to capture the overall trends of the velocity profiles, it underpredicts the velocity deficit at $X/D = 1.4$ and
beyond. The experimental results indicate that the wake centre is lower than predicted in CACTUS, though quantifying this
displacement is challenging due to the limited number of measured heights in the experiments. The vortical structures observed
in each plane are generally similar between CACTUS and the experiments, except for the shed vortex, which is visible at
$X/D = 0.5$ in the experiments but not in CACTUS. Further downstream, the vortical structures primarily consist of trailing
vortices from the upper and lower blade tips. However, CACTUS appears to overpredict the vortex strengths, consistent with
the discussion in Section 4.1. This discrepancy arises because, in the experimental results, the vortices exhibit dissipation,
whereas in CACTUS, vortices generated in previous cycles remain dominant in the flow.

(a) $\beta = 0°$ (baseline)

(b) $\beta = -10°$

(c) $\beta = 10°$

**Figure 8.** Normalised streamwise velocity ($u/U_\infty$) contours of the X-Rotor with pitch offset $\beta = 0°, -10°, 10°$ respectively at downstream locations $X/D = 0.5, 0.7, 1, 1.4,$ and $1.6$ of the CACTUS (above) and experiments (below). The black dash lines indicate the projected frontal area of the rotor on the corresponding plane. The x-axis is magnified to enhance visibility.

When pitched to $\beta = -10°$, the lateral thrust of the rotor increases in magnitude, inducing a laterally inward flow into the wake (Huang et al., 2023; Giri Ajay and Simao Ferreira, 2024). This results in wake inflow from the sides and vertical outflow, a pattern observed in both the experiments and CACTUS. However, differences arise in the velocity magnitudes and wake
shape. CACTUS predicts stronger lateral flow on the windward side compared to the experiment, causing the wake to contract





(a) $\beta = 0°$ (baseline)

(b) $\beta = -10°$

(c) $\beta = 10°$

**Figure 9.** Normalised streamwise vorticity ($\omega_x D/U_\infty$) contours of the X-Rotor with pitch offset $\beta = -10°, 0°, 10°$ respectively at downstream locations $X/D = 0.5, 0.7, 1, 1.4,$ and $1.6$ of the CACTUS (above) and experiments (below). The black dash lines indicate the projected frontal area of the rotor on the corresponding plane. The x-axis is magnified to enhance visibility.

more rapidly in this region. Additionally, CACTUS shows the wake starting to exit the top of the rotor area at $X/D = 0.7$, whereas this behaviour is not yet observed in the experiments. The discrepancy increases further downstream, attributed to the stronger tip-vortices predicted by CACTUS, which accelerate wake recovery compared to the experiments. Moreover, CACTUS predicts higher velocity deficits, suggesting a greater streamwise thrust than observed in the experiments. The large





pitch angles in this configuration cause the turbine to operate at extreme angles of attack, well beyond stall, which significantly influences CACTUS predictions since it relies on polar data for load estimation. Bensason et al. (2024) documented substantial unsteady flow separation and turbulence on the windward side at this pitch setting, explaining the stronger windward tip-vortex in CACTUS that leads to a more contracted wake. Additionally, the difference in circulation between the upper and lower blades results in a root vortex predicted by CACTUS. This vortex is visible in the windward root section and gradually
moves upward downstream. Furthermore, CACTUS vortices do not dissipate as quickly as those in the experiments, further contributing to the observed differences.

    In the $\beta = 10°$ case, the opposite wake behaviour is observed—laterally exiting through the upper half of the rotor area while freestream air enters vertically from above. The experiments indicate significant wake deflection starting at $X/D = 0.5$, which is greater than the values predicted by CACTUS. Additionally, CACTUS underpredicts the velocity deficit within the rotor
compared to the experiments until $X/D = 1$, beyond which most of the experimental wake exits the captured field of view. These discrepancies are primarily attributed to the vorticity distribution arising from the load distribution. In the experiments, a dominant windward tip-vortex facilitates freestream inflow from above. Meanwhile, the weaker vortices dissipate quickly, preventing them from significantly influencing the wake. In contrast, CACTUS, lacking viscous dissipation, produces numerous vortices near the windward tip, which intensify further downstream. This incorrect vortex representation leads to inaccurate
velocity predictions, preventing CACTUS from capturing the downward movement of the wake. Interestingly, the wake in the lower half of the X-Rotor shifts laterally in the opposite direction to the upper half. This is due to the formation of a root vortex that induces lateral forcing on the lower half. Once again, differences in vortex dissipation contribute to this effect—while the experimental data show a dominant tip-vortex influencing vertical freestream influx, CACTUS instead predicts an accumulation of vortices near the upper half, altering the wake dynamics.

A quantitative analysis of the streamwise velocity profiles along the height of the X-Rotor is presented in Figure 10. To ensure consistency, we selected the same lateral locations as in the discussion from Section 4.1.

    In the leeward slice, the velocity profile from the experiment in the baseline case is generally well represented by CACTUS, except at $X/D = 1.4$ and beyond, where discrepancies become more pronounced. This aligns with the earlier observation that CACTUS predicts the wake centre to be positioned higher than in the experimental results. For $\beta = -10°$, a significant
difference emerges between the predicted and measured values, with the experiments showing much higher velocities than CACTUS. This discrepancy increases further downstream, reaching up to 50% along the height. The primary cause of this difference is the previously discussed overprediction of tip-vortex strength in CACTUS. At $\beta = 10°$, CACTUS predictions also deviate from experimental values. While the velocity profile aligns with the experiment over a small height range in certain planes ($X/D = 1$ and 1.6), larger deviations occur near the root. This discrepancy is primarily due to the stronger
leeward vortex predicted by CACTUS compared to the weaker vortex observed in the experiments. However, the differences in this pitch case at the leeward slice are generally smaller than those seen in the $\beta = -10°$ case.

    A similar pattern is observed in the windward slice—while the streamwise velocity profile is predicted relatively accurately in the baseline case, the pitched cases exhibit significant inconsistencies with the experimental profiles. These discrepancies are primarily attributed to differences in vortex strength predicted by CACTUS, which depend on the vortex core sizes selected



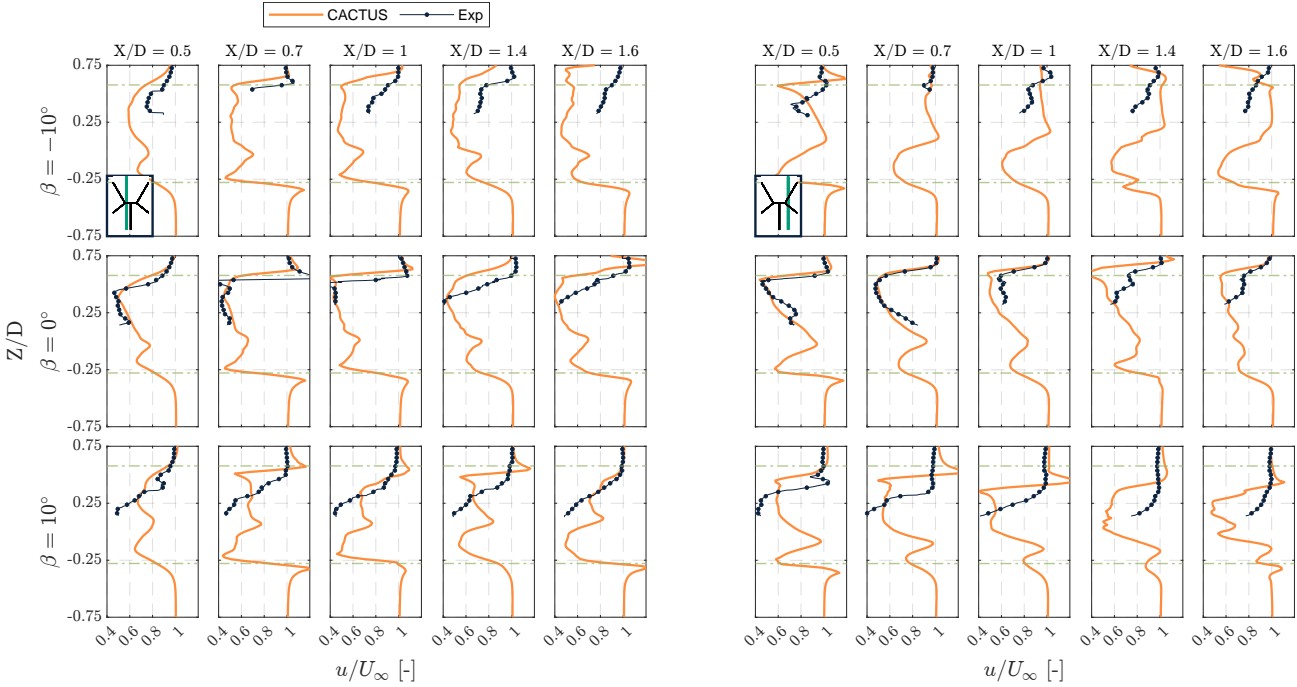

**Figure 10.** Comparison of the streamwise velocity $u/U_\infty$ slices along the height of the X-Rotor at lateral position $Y/D = -0.125$ (left) and 0.25 (right) at discrete streamwise locations $X/D$ = [0.5, 0.7, 1, 1.4, 1.6] at pitch offsets $\beta = [-10°, 0°, 10°]$. The inset figure in the first tile is presented to indicate the location of the slice with respect to the rotor - the green line is the $Y/D$ slice along the height of the rotor.

for the simulation. Moreover, CACTUS struggles to accurately capture the increase in circulation induced by pitch offsets, as its predictions rely heavily on the input polars. Given that the static stall angle for these polars is $\alpha_{ss} = 9.1°$, pitching the airfoils by $\beta = 10°$ introduces additional uncertainty in airfoil performance, particularly at these low Reynolds numbers. Additionally, flow curvature effects could potentially mitigate these differences by improving the vortex representation in the pitched cases, but their influence on a coned blade remains unclear. Since the wake characteristics are directly linked to blade forces, a comparison of the predicted and experimental blade loads would provide further insight. However, blade load data is unavailable for this dataset.

Overall, CACTUS appears less effective in capturing the near-wake flowfield of the X-Rotor without pitch offsets. However, the general trends and behaviour of the wake profiles predicted by CACTUS remain valuable for future research.

### 4.3   Influence of cone angle on the velocity field

As the CACTUS model represents the flowfield of the case without pitch offsets well, we can consider its predictions on the flowfield to be valid.

In our previous work (Giri Ajay et al., 2024), we introduced the concept of vertical induction generated by the coned blades of the X-Rotor, both with and without pitch offsets. In this section, we extend that discussion by presenting the spanwise normal



force distribution and spanwise angle of attack distribution as a function of azimuth in Figure 11. Additionally, we provide the
blade-integrated forces to offer insight into the overall load contribution of the rotor. In both figures, we focus on the B1 alone,
rather than the entire rotor, to facilitate a more detailed discussion of the downwind half of the rotation.

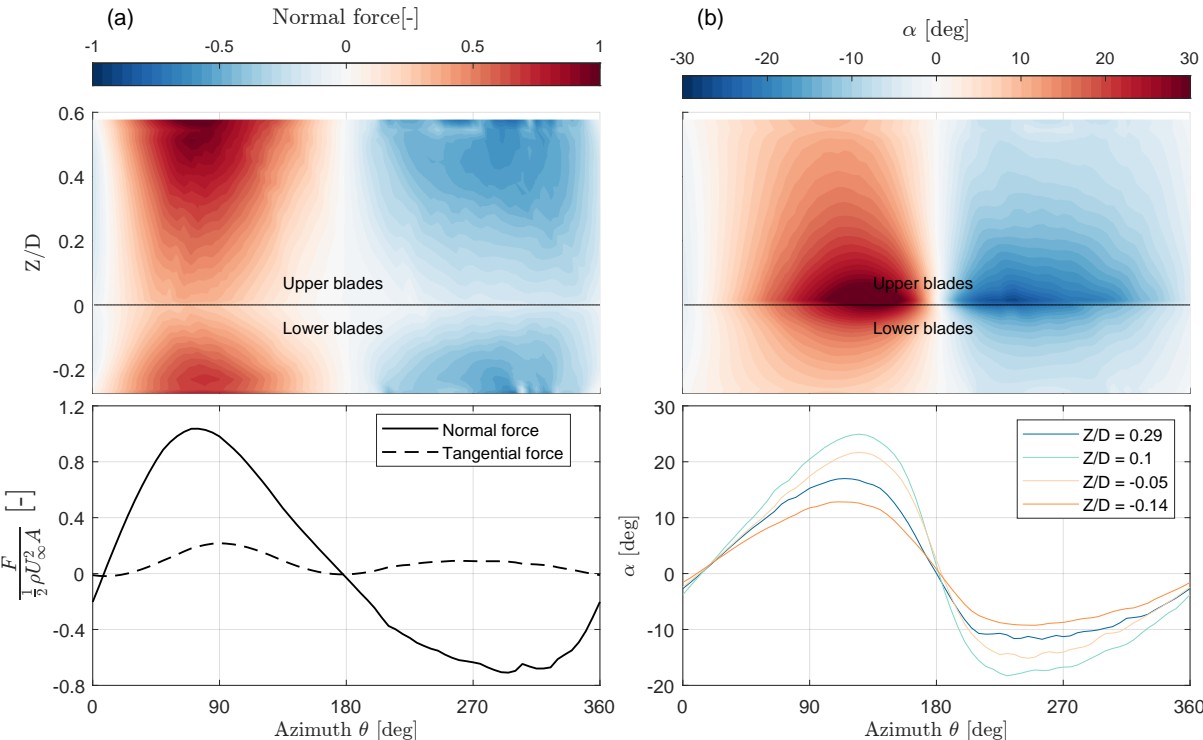

**Figure 11.** (a) Spanwise distribution of normal blade force as a function of height $Z/D$ and azimuth $\theta$ in the top tile and blade integrated
normal and tangential force as a function of azimuth in the bottom tile. Positive normal force is away from the axis of rotation and vice versa.
Forces are integrated along both upper and lower halves of B1. (b) Angle of attack $\alpha$ variation along height and $\theta$ in the top tile, and bottom
tile shows $\alpha$ at discrete locations along height.

In general, the forces in the upwind half are larger than in the downwind half, reaching their maximum near the most upwind
and downwind positions. This behaviour is consistent with existing VAWT normal force profiles for non-pitched blades and
aligns with our previous findings. The spanwise distribution reveals that forces are highest near the tip and lowest near the root
section, which is expected as the local inflow velocity decreases closer to the root. However, some artefacts at the upper and
lower tips indicate a local increase in load. Lifting-line methods consider induction only along a line at the centre of pressure
(quarter-chord in this case), neglecting chordwise distribution effects. This often results in an overestimation of loading near
the tip (Sørensen et al., 2016). As CACTUS does not currently incorporate a correction model for this issue, and implementing
one is beyond the scope of this study, we neglect this local spike as it is not critical to our analysis. Consequently, the peak
load along the span occurs around $Z/D = 0.5$ in the upper half and $Z/D = -0.23$ in the lower half, with a gradual reduction
in magnitude towards the tips—an expected characteristic of a finite blade span representation in CACTUS. Interestingly, in



the downwind half between $\theta = 270°$ and $330°$, the forces fluctuate near the tips while remaining relatively stable elsewhere. This is attributed to blade–vortex interactions, where the blades pass through tip vortices shed during the previous cycle. This interaction is also evident in the integrated forces, where a small spike is observed in the same azimuthal range. Our previous study (Giri Ajay et al., 2024), using both another free-wake vortex model and a blade-resolved URANS approach, predicted a similar phenomenon. Since the vortices exhibit minimal vertical convection, the rest of the downwind region remains largely unaffected by blade–vortex interactions, a consequence of the coned blade geometry.

The angles of attack, $\alpha$, decrease from the root toward the tips, reflecting the variation in local blade section rotational velocity along the height. Given that the static stall angle is $\alpha_{ss} = 9.1°$, most of the blade operates in either post-stall or deep-stall conditions between $\theta = 90°$ and $180°$. The blade sections near the root ($Z/D = 0.1$ and $Z/D = -0.05$) experience deep-stall conditions, with peak $\alpha$ reaching approximately $25°$ in the upwind half and $19°$ in the downwind half. This aligns with the observed low normal forces at these blade sections.

To understand the implications of the spanwise variation of normal forces on the flowfield, we present the time-averaged vertical component of velocity $\bar{w}/U$ at different height sections as viewed from above in Figure 12.

At $Z/D = 0.29$, we primarily observe downwash within the rotor volume, driven by the vertical component of the normal forces from the upper blades. Interestingly, downstream of the rotor, downwash persists within the region bounded by the local diameter, whereas upwash dominates outside this region ($Z/D > |0.2|$). Moreover, this upwash is stronger on the windward side than on the leeward side, indicating the influence of the tip vortices. The windward tip vortex, over a full cycle, is stronger than the leeward tip vortex. This behaviour closely aligns with the observations of Bensason et al. (2024), who reported a similar wake structure. Specifically, at $\beta = 0°$ in their study, the windward tip vortex induced upwash on the windward side at this plane.

At $Z/D = 0.1$, the downwash is stronger than at the plane above and extends further into the wake. This is due to the cumulative contribution of downwash from the blade sections positioned above this plane. As a result, the downwash reaches further downstream while remaining laterally confined within the bounds of the local diameter. The influence of the upper tip vortices is reduced at this plane, as it is farther from the tips, which is evident from the lower intensity of the red regions.

In the lower half of the rotor at $Z/D = -0.05$, the region within the local diameter exhibits minimal upwash. This is because the downwash generated by the upper half counteracts the effect of the higher cone angle in the lower half, which results from the greater loads produced by the upper blades. Additionally, this plane captures the cumulative upwash generated by the entire lower half. However, the wake predominantly features a strong downwash—likely an effect of the lower tip vortex—but this influence does not extend significantly beyond $X/D = 1$. Around $Y/D = -0.25$, we observe a region of upwash that extends from $X/D = 0.5$ to $1.5$. This is due to the cumulative effect of the shed vortices from both the upper and lower blades in the leeward region, as observed from the angle of attack plots. Near the root, these shed vortices induce a resultant upwash due to the difference in cone angles between the upper and lower blades.

Finally, at $Z/D = -0.14$, the region inside the local diameter is primarily characterised by upwash. However, as observed in the previous plane, the wake remains dominated by downwash. Once again, the windward side experiences stronger downwash than the leeward side, attributed to the windward tip vortex. Since the vortices shed by the lower blades rotate in the opposite



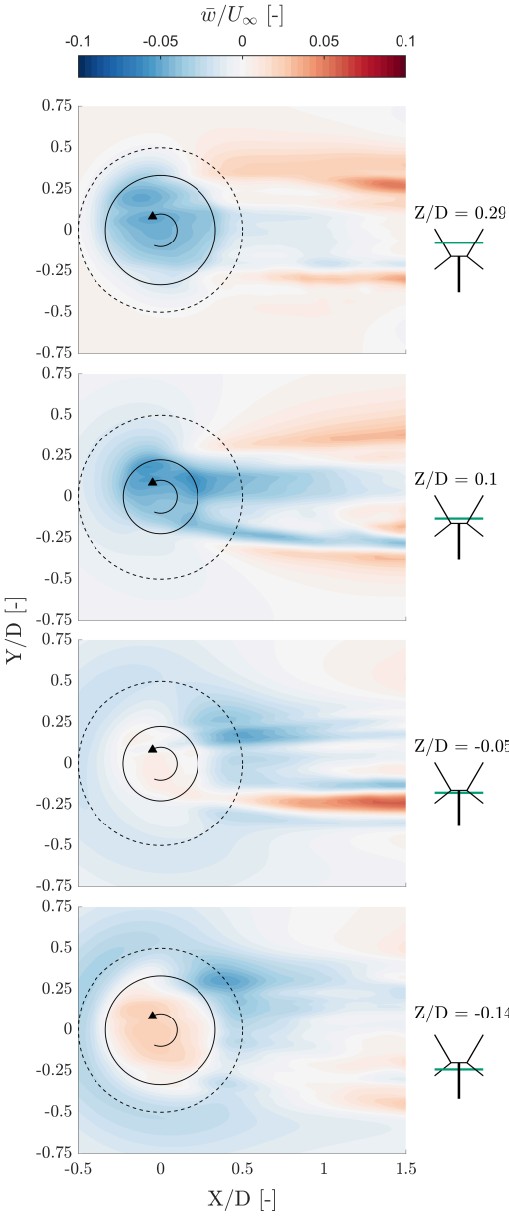

**Figure 12.** Time-averaged vertical component of the velocity field of the rotor at sections along the height of the rotor at $Z/D$ = 0.29, 0.1, -0.05, and -0.14 as viewed from the top. The inset figure shows the locations of the planes along the height of the X-Rotor graphically. The dashed black line indicates the tip-diameter of the X-Rotor and the solid black line indicates the local diameter corresponding to the height. Red indicates upwash (out of the plane) and blue indicates downwash (into the plane).

direction to those from the upper blades, this behaviour is expected. Additionally, the small region of upwash in the wake is likely caused by the leeward tip vortex.



Overall, the cone angle significantly influences the flowfield both inside and outside the rotor, and this effect is further
amplified by the interaction of tip vortices when the blades are pitched.

## 5   Conclusions

We conducted a validation study of CACTUS, a free-wake vortex tool, against wind tunnel PIV measurements for a scaled X-
Rotor VAWT, considering cases with and without blade pitch offsets. Additionally, we examined the influence of coned blades
on the near wake in the absence of pitch offsets, highlighting the significance of vertical induction for such VAWT geometries.
The results from CACTUS were compared with phase-locked stereo PIV measurements taken at different azimuths.

Our findings indicate that CACTUS effectively represents the flowfield within the X-Rotor volume and in the very near
wake for cases without blade pitch offsets. The model captures the trends and flow features well, though some discrepancies
in velocity magnitude arise due to the choice of model setup parameters. Further inaccuracies stem from aeroelastic effects
present in the experiments.

The spanwise distribution of blade forces shows a reduction in magnitude towards the root, as expected, due to the decrease
in local inflow. Since the forces are more pronounced near the tips, it is evident that the coned blades significantly influence
the flowfield by inducing vertical velocity.

Examining the effect of cone angle on the downstream flow, we found that a local downwash is generated within the rotor
volume by the upper blades, while the lower blades produce an opposite effect. Additionally, the turbine wake exhibits sub-
stantial vertical flow (ranging from 5–10% of the freestream) even at the root sections, where blade forces are relatively small.
This vertical velocity field is attributed to the rotor's tip vortices, which are expected to become more pronounced with pitch
offsets as the blade loads are altered.

When comparing cases with blade pitch offsets, CACTUS exhibited significant discrepancies from the PIV results. While
the general wake behaviour was captured, the rate of wake advection was misrepresented. These differences were primarily
attributed to variations in tip-vortex size predicted by CACTUS, which stem from the chosen model setup parameters. Con-
sequently, the discrepancies observed in the case without pitch offsets were amplified when pitch offsets were introduced, as
the wake of a VAWT with blade pitch offsets is strongly influenced by tip vortices. Quantitative analyses revealed that in some
instances, CACTUS predictions deviated by up to 50% from the experimental results. These differences also stem from the
use of airfoil polars, which may not accurately represent the load profile at low Reynolds numbers, particularly at the large
pitch offsets considered in this study. While careful tuning of the CACTUS setup for the specific case could help reduce these
discrepancies, this potential reduction could not be quantified in the present study.

To conclude, CACTUS is an excellent tool for simulating the aerodynamics of the X-Rotor in the baseline case, accurately
capturing the flowfield within the rotor volume and the very near wake. However, for cases with blade pitch offsets, a different
modelling approach is needed to better predict the wake flow features, particularly in terms of vortex dissipation and evolution.





*Data availability.* The data that support the findings of this study will be made openly available in 4TU ResearchData.

## Appendix A:  Sensitivity study: influence of vortex core size, dynamic stall, and the choice of Reynolds numbers

Initially, we started with an azimuthal discretization of 144 elements, which corresponds to a resolution of $\Delta\theta = 2.5°$ and no dynamic stall. All vortex core radius were set to 100% of the chord, and using the other settings presented in Table 1. We shall call this case the 'default' case. We wanted to address the sensitivity in the following points: (1) vortex core radius choice,
(2) with and without dynamic stall model, and (3) lift $C_L$ and drag $C_D$ coefficients. While the first couple of points are more standard, we wanted to identify how sensitive are the results to the chosen airfoil polars - especially in the flowfield inside the rotor.

To address this, we first analyse the normalised streamwise velocity ($u/U_\infty$) contours at azimuth $\theta = 0°$ for these cases, available in Figure A1.

We observe that going from the default case to a lower vortex core radius does not seem to alter the flowfield visibly, but it introduces quite a lot of instability in the simulations which are reflected near the bottom tip. With the introduction of dynamic stall, some flow features that were not captured in the default case are now visible here, and they agree quite well with the flowfield from the PIV results. With an artificial increase in $C_L$ and $C_D$ of 15%, there appears to be some instability again, as the vortex core size for that case is small with respect to the vorticity strength it offers. However, we see significant
velocity magnitude changes between the default case and this inside the rotor area, even at the most upwind plane. Overall, we understand that any changes to lift and drag still significantly affects the flowfield within the volume of the rotor, and that including dynamic stall more accurately represents the flowfield of the X-Rotor.

Moving on towards representation of blade forces, we present the integrated blade forces between the default case, the case with dynamic stall, and the lowered bound and trailing vortex core radius in Figure A2.

Between the default and reduced vortex core size cases, we see minor differences in the load - notably in the downwind half where blade vortex interaction is expected (around $\theta = 300°$). However, significant difference exists between the dynamic stall case and the default case. We see the normal load peak being larger and occuring slightly later, indicating the delay introduced by the dynamic stall model. In the downwind half, we do not see significant difference between the two. But notably, we see spiky behaviour in the loads for the dynamic stall case, and it is likely related to the fine azimuthal resolution chosen in this
case.

To further elaborate this, we present the spanwise distribution of the forces for the same cases in Figure A3. One change we made is to reduce the azimuthal resolution to $\Delta\theta = 5°$ for the dynamic stall case to check its impact on the non-smooth behaviour of the force profile.

We see that there is very minor differences between the first two tiles, as observed in the integrated forces. But additionally,
we notice that there is some spiky artefacts in these two tiles around $Z/D = 0.9$ and exists in the upwind and downwind windward regions. Whereas on the third tile, after reducing the azimuthal resolution, we do not see the spiky behaviour and

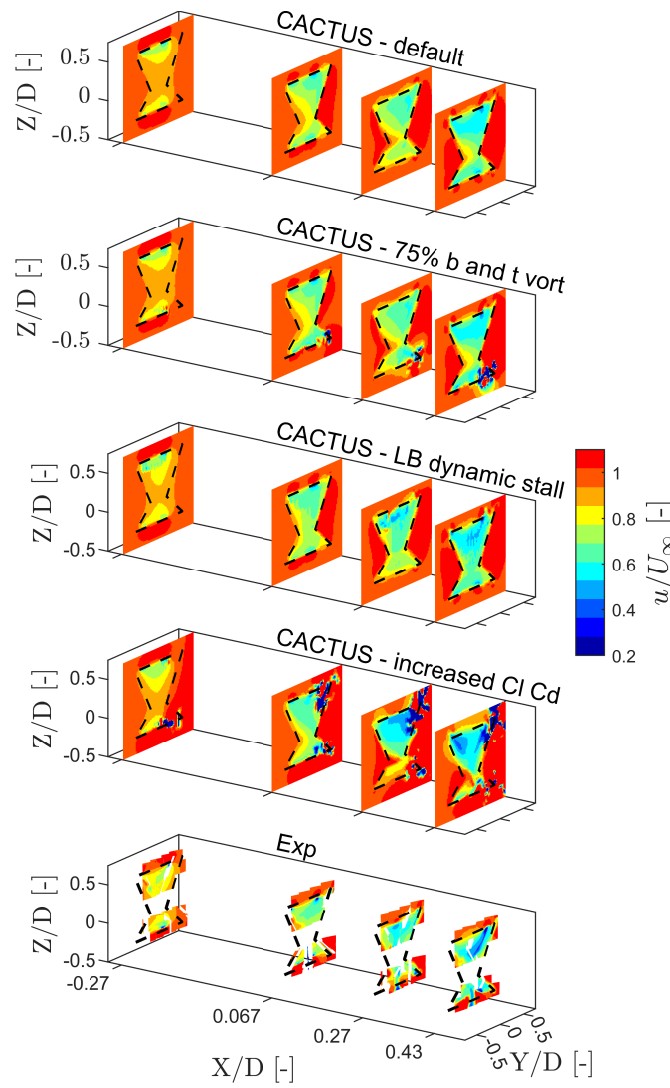

**Figure A1.** Normalised streamwise velocity ($u/U_\infty$) contours of the X-Rotor at azimuth $\theta = 0°$ at downstream locations $X/D$ = -0.27, 0.067, 0.27, 0.43 where $D$ is the rotor diameter for different input conditions. The first tile is the default case, the second is the case where the bound and trailing vortices are reduced to 75% of chord, third tile shows the use of dynamic stall, fourth tile shows a case with an artifical increase in $C_L$ and $C_D$, and the last tile shows the experimental results. The black dash lines indicate the projected frontal area of the rotor on the corresponding plane. All spatial coordinates are normalised the diameters $D$. The x-axis is magnified to enhance visibility.

we observe smoother profiles. Furthermore, the dynamic stall model shows higher loads near the roots which the default case is not able to represent.

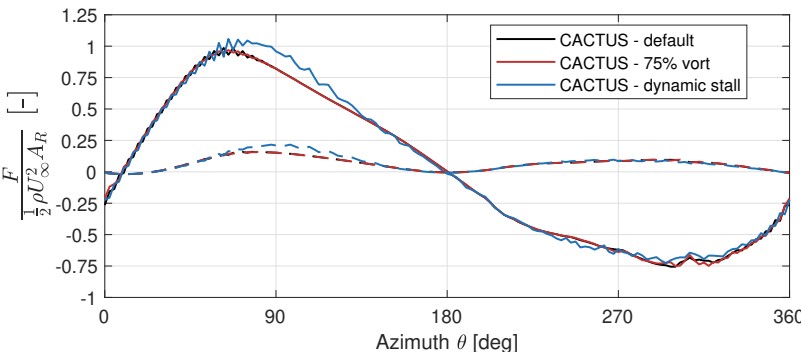

**Figure A2.** Integrated blade forces $F$ as a function of azimuth $\theta$. The solid lines represent the normal force and tangential force is represented by the dashed lines. Positive normal force is away from the axis of rotation and vice versa. Forces are integrated along both upper and lower halves of B1.

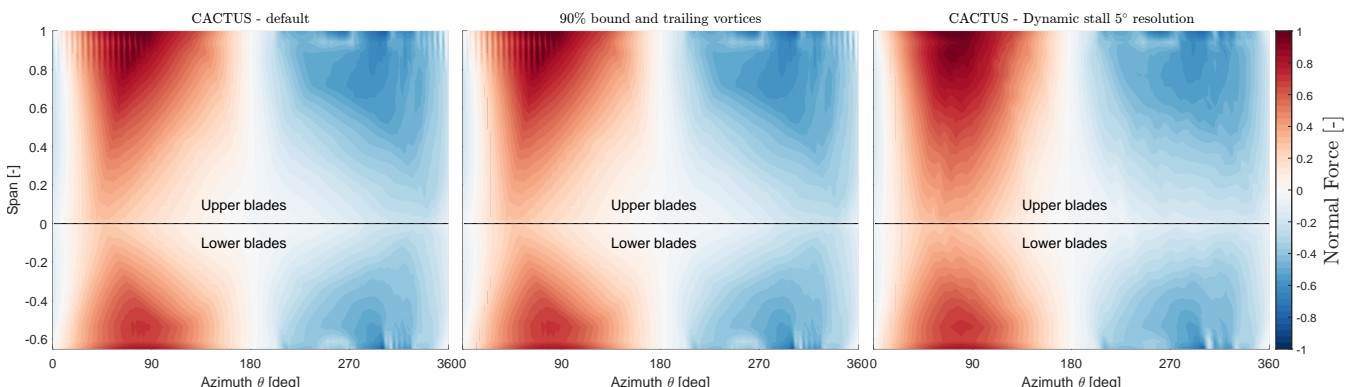

**Figure A3.** Spanwise distribution of normal blade forces $F_N$ as a function of azimuth $\theta$. The solid lines represent the normal force and tangential force is represented by the dashed lines. Positive normal force is away from the axis of rotation and vice versa. Forces are integrated along both upper and lower halves of B1.

Therefore, given these benefits of using dynamic stall in this analysis and the computational cost reduction by reducing the azimuthal resolution, we choose to operate with the Leishman-Beddoes dynamic stall implementation in CACTUS and proceed to use an azimuthal discretisation of 72 instead of 144 - which corresponds to a resolution of $\Delta\theta = 5°$.

*Author contributions.* AGA did the main research and analysis for the numerical results from the vortex models, and wrote the paper. DB contributed towards conducting the experiments, assisting with the results and discussion, while writing the section about the experimental setup. DDT guided the analysis and the model setup, helped with the model setup and the sensitivity studies, and helped with reviewing and editing the work. The paper was revised and improved by all authors.





*Competing interests.* The authors declare that we have no competing interests.

*Acknowledgements.* We wish to thank the European Union's Horizon 2020 research and innovation programme for funding this research under grant agreement no. 101007135 as part of the XROTOR project (https://doi.org/10.3030/101007135). We would also like to acknowledge Prof. Carlos Ferreira's efforts in conceptualising this study and enabling the existence of this work.



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
