# Peer review of "Validation of the near-wake of a scaled X-Rotor vertical-axis wind turbine predicted by a free-wake vortex model"

_Wind Energy Science, 2025_

## Author Comment (AC1)

**Manuscript ID: WES-2025-54**

Validation of the near-wake of a scaled X-Rotor vertical-axis wind turbine predicted by a free-wake vortex model

**Adhyanth Giri Ajay, David Bensason, and Delphine De Tavernier**

**April 2025**

The authors would like to thank the reviewers for their time, and valuable comments. Their inputs significantly improved the quality of the paper. The comments of each reviewer are addressed separately. The explanation for each question/comment is marked in blue while the actual changes in the manuscript are marked in red with the updated line numbers.

**Reviewer 1**

1. Comment (line 2 of the abstract): The phrase "to minimise the levelised cost of energy" is somewhat broad. If possible, please clarify in one sentence how the proposed X-Rotor specifically contributes to lowering the LCOE (e.g., through reduced manufacturing costs, improved efficiency, easier maintenance, etc.).

Thank you for highlighting it. We mentioned it in the introduction section. The X-Rotor specifically aims at reducing the capital and operational/maintenance expenditures through its design. We shall adjust the abstract to reflect this.

As part of this interest, a novel X-shaped VAWT (X-Rotor) has been proposed to minimise the levelised cost of energy by minimising capital and operational expenditures.

2. Comment (line 36, Introduction): It would be helpful to clarify whether the study by Morgan et al. (2025) also employed a BEM-based approach, and whether it included a quantitative comparison of power coefficients between the classical and X-type rotor configurations. This information would support the claim of performance benefits associated with the X-type design.

Thanks once again. Morgan et al. (2025) used a 2D Actuator cylinder approach to come to the conclusion. Their work did not focus on the X-type rotor configuration, but rather showcased the strengths of having a coned blade geometry through various inclination angles. Therefore, their study is not a standard comparison of H-type to the X-type design over the same rated power or frontal area.

Later, Morgan et al. (2025) demonstrated the power gains achieved by coned blades compared to non-coned blades for a given blade span using a 2D Actuator Cylinder approach.

3. Comment (Section 2.1): The manuscript mentions that the scaled model was derived from a large turbine design. Please clarify whether the scaling was purely geometric, or if similarity parameters (e.g., Reynolds number, tip-speed ratio) were also considered in the design process. Additionally, it would be useful to specify whether the blades were clean or if any flow tripping devices were used to promote separation. Finally, please provide details on the blade mounting configuration — specifically, was the mounting point located at the quarter-chord (c/4) position from the leading edge?

The scaling was purely geometric based on the diameter. The blades were clean and did not have any taper or any vortex generators. The chord-mounting point was c/2. In the paragraph below, please find the bold

texts to highlight the changes we made. The paragraph is also modified to account for the comments from Reviewer #2 as well.

The test geometry is a **purely** geometrically scaled model of the full-size primary rotor, reduced by a factor of  $\frac{1}{100}$ . The top and bottom blades have a tip diameter of D = 1.5 m and cone angles of 30° and 50°, respectively, resulting in upper and lower spans of 1 m and 0.65 m, respectively. Aside from scaling, the primary difference between the full-scale turbine and the scaled model is that the latter consists of four straight NACA0021 airfoils with a constant chord of c = 0.075 m, attached to a stiff crossbeam of length 0.5 m with the same profile and chord. The blades are clean, without any vortex generators and are mounted at c/2. The rotor is supported by a tower with a diameter of 0.06 m. The model operates at a constant tip-speed ratio  $\lambda = 4.0$  at  $U_{\infty} = 4$  m/s, yielding a chord-based Reynolds number of  $Re_c = 8.1 \times 10^4$  at the tip. The operating conditions are determined to obtain a thrust coefficient to be as close as possible to the optimal value of  $C_T = 0.7$ , without compromising the structural integrity of the rotor.

4. Comment (line 153): The authors state that the simulations were run for 10 revolutions to ensure convergence. Please clarify whether 10 revolutions represent the total simulated time. Is this duration sufficient to obtain both converged blade loads and a fully developed wake structure?

We present to you a plot on the convergence of the power and thrust with respect to the number of revolutions in Figure 1. Beyond the 8th revolution, the errors begin to oscillate between the 3rd and the 4th order of precision. This, in our opinion, is sufficient enough to capture the lower order effects observed in this study, as we are not modelling turbulence. This has been added to Appendix A, as a short subsection.

Figure 1: Convergence plot of power and thrust. The Y-axes are in logarithmic scale

5. Comment (starting around line 167): I am not sure whether the figure numbering is consistent starting from line 167. The paragraph beginning at line 162 discusses Figs. 5 and 6, but, if I'm reading correctly, there appears to be no reference to Fig. 7, and the text jumps directly to Fig. 9. Moreover, Fig. 9 already refers to the case with pitched blades, suggesting that an intermediate figure (possibly Fig. 7 or 8) might be missing or misnumbered.

Thank you for pointing it out. We have now fixed the Figure numbers. You can find the reflected changes in the marked up version of the manuscript.

Overall, the velocity predictions within the rotor volume show good agreement with the experimental results, as observed in **Figure 5**, except for the wake of the tower . . .

A similar trend is observed in the vortex structures (Figure 6), . . .

6. Comment (around line 175): The authors rightly note that the numerical approach fails in certain regions. I believe two factors may contribute to this discrepancy: the accuracy of the airfoil characteristics at such low Reynolds numbers, and potential limitations of the dynamic model used. A brief discussion of these aspects would strengthen the interpretation of the results.

Thanks for your suggestions. We did consider the airfoil characteristics at low Reynolds numbers and mentioned it in the conclusions, however neglected to highlight it as an obvious answer to all our validation cases present here. We already referred to the Melani et al. (2019) while choosing our polars, citing significant variation even at the Reynolds number of our choice.

With regard to the dynamic stall model, we could not compare multiple dynamic stall models to be able to scientifically address the limitations of using the Leishman-Beddoes model, specifically for the X-Rotor. While we did try to run a case with the Boeing-Vertol model (pre-packaged with CACTUS) it resulted in diverging solutions and we did not present them here.

Notably, as highlighted in Section 3.2, the difference could also arise from the polars due to the inconsistency shown between experimental and numerical lift coefficient profiles. This holds true for the rest of the results discussed in this study.

**Reviewer 2**

1. There is significant uncertainty in the polar data and overall aerodynamic modeling, particularly in both pre- and post-stall regimes. Assuming similar airfoil behavior at Re = 80k and 150k is a strong simplification, especially considering the chosen airfoil and the experimental turbulence intensity. Using XFOIL with Ncrit = 8 is also questionable, as the experimental TI suggests a lower Ncrit  $\approx 4$ . This likely introduces notable errors in force predictions and may account for discrepancies with the experiments such as the reduced wake expansion. Additionally, the treatment of post-stall behavior is unclear—was any extrapolation method used?

Thank you for your very thorough insight into the paper. We shall address your comment in reverse order of your comments, as each point is dependent on the other. We performed a Viterna extrapolation Viterna and Janetzke (1982) to obtain post-stall behaviour of the polars. We shall mention this in the numerical setup section. Regarding the chosen value of Ncrit, the reviewer is indeed correct that using an Ncrit = 4 would be more suitable based on the experimental TI. This was a critical oversight on our part. We further agree with the reviewer that performing the simulations at Re = 80k as opposed to Re = 150k would improve the reliability of the results, especially with the updated Ncrit value. Therefore, we have repeated our study with Re = 80k and an Ncrit = 4 for all the pitch cases. After a very thorough inspection, we noticed that the results do not change significantly at all - often remaining extremely similar to our previous result. This is attributed to the very close agreement between the load profiles generated by using the two polars (see Figure 5 and Figure 3 below). Therefore, there appears to be no significant changes to our observation throughout this study, however we have used a more reasonable polar for the validation.

Figure 2: Integrated blade forces F as a function of azimuth  $\theta$ . The solid lines represent the normal force and tangential force is represented by the dashed lines.

Figure 3: Spanwise distribution of normal blade forces  $F_N$  as a function of azimuth  $\theta$ .

2. Flow curvature, which becomes significant for blade sections with chord-to-radius ratios c/R > 0.1, is ignored in the simulations. This is a critical omission, given that most of the blade operates beyond this limit and flow curvature has a strong effect on the lift characteristics at low angles of attack

Yes, the flow curvature model is more pertinent for the X-Rotor than the typical H-Darrieus VAWT. This is because in the X-Rotor, as the reviewer pointed out, most of the blade does operates beyond this limit (especially near the root). While we would definitely have used the flow curvature correction for typical VAWTs with similar chord-to-radius ratios, we omitted it in this study for two critical reasons.

Primarily, as mentioned in the methodology, it would become exceedingly difficult to isolate the differences observed between the predicted values from CACTUS and the experiments with the flow-curvature on for the X-Rotor geometry. Historically the flow curvature has been shown to increase the angle of attack in the upwind half and lowering it in the downwind half (Goude, 2012). This has not yet been verified for the X-Rotor geometry, with a prominent spanwise variation of relative velocity.

Secondly, the flow-curvature model is not built into CACTUS. Any attempts of modifying the source code of the solver is outside the scope of this manuscript. Furthermore, attempting to pre-emptively correct the polars without calculating the relative velocities would invite unwarranted errors in the free-wake vortex model. We believe including the correction is essential, but is not crucial in the context of validation at these low Reynolds number regimes, as the uncertainty of the polars would be the primary point of contention. We have expanded on this in Section 3.3.

Unrelated to this study, in our on-going validation study between an actuator line method (turbinesFoam) and experiments for the X-Rotor geometry, we performed a sensitivity study between the regularisation kernel ( $\epsilon$ ), dynamic stall, and flow curvature (see Figure 4 below). While we definitely need to the tune the inputs more to match the velocity values, we can see that the flow curvature model causes the wake shape to deviate away from the experiments. Without the flow curvature, we can see the wake shape predicted is closer to the experiments. So indeed, modelling the flow curvature is critical. But, its limitations for this geometry are unclear at present as it seems to have a detrimental effect towards the accuracy of the wake. However, due to the limitations of CACTUS, we are unable to provide any scientific insight into the effect of flow curvature in this study.

---

## Referee Report (RR1)

This manuscript presents a comparison between a free-wake vortex model (CACTUS) and stereoscopic PIV measurements for a scaled X-Rotor, with analysis covering both baseline and pitched-blade cases. The study offers insight into wake development and 3D flow structures introduced by blade coning. However, it lacks sufficient validation and verification of CACTUS, especially in terms of its ability to predict aerodynamic loads. Most discrepancies between simulation and experiment are attributed to the vortex core size, without adequately exploring other potential causes.

1. Choosing not to include the flow-curvature (virtual camber) effect to "avoid complicating error attribution" is questionable—excluding a known aerodynamic influence makes it harder, not easier, to understand discrepancies. A practical alternative would be to preprocess the airfoil polars by incorporating the effective camber caused by flow curvature under typical VAWT conditions. These adjusted polars could be used as CACTUS input, offering insight into the impact of virtual camber without modifying the code itself.

2. The authors attribute wake discrepancies—particularly at $X/D = 0.43$—to the vortex core size, supported by a reference to an actuator line method (ALM) study. While there is a conceptual similarity between force smearing in ALM and vortex core effects in lifting-line theory, the two methods are fundamentally different, and this comparison is not directly applicable. Concluding that the discrepancies stem from vortex convection assumes the aerodynamic loads are accurate—something not demonstrated in the paper. The brief mention of uncertainties due to airfoil polars is more relevant and deserves further exploration before focusing solely on the vortex core.

3. In the pitched case, where discrepancies grow more significant, the authors again attribute the issue to the vortex model while overlooking the potential role of load misrepresentation, such as inaccurate polars. This is a missed opportunity for a more balanced analysis.

4. Finally, the sentence in the abstract—"Results indicate that CACTUS effectively replicates the flowfield within the rotor volume and the very near wake when no pitch offsets are applied"—overstates the level of agreement. While results align with experiments in some areas, noticeable discrepancies remain. A more measured phrasing would better reflect the findings.

Based on these concerns, I recommend **major revision** before the manuscript can be considered for publication.

---

## Author Response (AR2)

**Manuscript ID: WES-2025-54**
**Validation of the near-wake of a scaled X-Rotor vertical-axis wind turbine predicted by a free-wake vortex model**

Adhyanth Giri Ajay, David Bensason, and Delphine De Tavernier

April 2025

The authors would like to thank all three reviewers for their time, and valuable comments. Their inputs further improved the quality of the paper, especially with the additional comments from Reviewer #3. The comments of each reviewer are addressed separately. The explanation for each question/comment is marked in blue while the actual changes in the manuscript are marked in red with the updated line numbers.

**Reviewer 2**

*1.Polar data: In the absence of reliable experimental measurements, the presented sensitivity analysis represents a significant improvement in addressing uncertainties in the input polar data. Could you please confirm whether the results shown in the manuscript have been updated to reflect this revised input?*

Thank you for the complement. We can indeed confirm that all the results have been updated to reflect this revised input. The differences between the updated results were extremely minor, as mentioned in the last Author Comments. This was due to the blade forces being roughly similar to the first version of the manuscript. We can demonstrate it below by showing a couple of velocity slices below from the first draft of the manuscript and the current version with the polars at an Ncrit of 4 and a Reynolds number 80K. Below, you can find the minor differences pointed out in streamwise velocity slices along the height for $Y/D$ = -0.125 for $\beta = 0°$ which can be most easily seen with the CACTUS results where the red arrow is pointing to at $X/D = 0.27$. The CACTUS results with the updated polars predict marginally higher velocities in the flowfield than before.

[Figure]

Figure 1: Streamwise velocity $u/U_\infty$ slices along the height of the X-Rotor at lateral position $Y/D = -0.125$ at $\theta = 0°$ and $\beta = 0°$. The X-axis is the streamwise velocity (it is cropped from the figures on the manuscript) On the left is the results from the first draft and on the right is the current version of the manuscript. The red arrow points to some discrepancies to easily notice them.

*2.Core radius setup: While the sensitivity analysis on the load predictions provides valuable insight, it does not fully justify the chosen value of the core radius—particularly given the study's focus on wake dynamics.*

*To strengthen this aspect, please also report the influence of the core radius on the mean wind speed and turbulence intensity in the near-wake velocity field.*

This is a very fair request. We have shown the influence on the core radius on the mean wind speed. With the free-wake vortex model, we are not resolving the turbulence in the flow - as there are no closure models used for it. Therefore, we have only provide the streamwise velocity contours in the appendix section. As you can see, there is very little difference in the flowfield between the chosen vortex core size as well as the 10% core size. Therefore, as there is no appreciable difference in the streamwise velocity profiles, we believe our choice of vortex core size does not affect the wake characteristics observed in the study.

Comparing the velocity fields with 10% vortex core size and the default at $\theta = 0°$ (Figure 2) shows minor impact in the flowfield that we consider the solver to be relatively independent of the vortex core size.

[Figure]

Figure 2: Normalised streamwise velocity ($u/U_\infty$) contours of the X-Rotor at azimuth $\theta = 0°$ at downstream locations $X/D$ = -0.27, 0.067, 0.27, 0.43 where $D$ is the rotor diameter for different input conditions. The first tile is the default case, the second tile shows the case with 10% vortex core size, and the last tile shows the experimental results. The black dash lines indicate the projected frontal area of the rotor on the corresponding plane. All spatial coordinates are normalised the diameters $D$. The x-axis is magnified to enhance visibility.

*3.Flow curvature: This is indeed a complex issue and likely beyond the scope of resolution within the current study. However, I recommend further revising the manuscript to acknowledge and discuss the limitations that this aspect introduces to your approach.*

Thank you. We agree that it is quite complex to be tackled within the current study. We have further elaborated the limitation of this model in the methodology section. We recommend reading the tracked changes to fully understand how this elaboration is done in conjunction with comment 1 of Reviewer #3.

We believe flow curvature model is essential for this geometry, but would bring uncertainty to the results when the behaviour of the rotor with and without flow-curvature has not been tested for this geometry. This is indeed a limitation in our approach as the lack of flow curvature would introduce differences in the blade forces and affect the near-wake due to the change in vortex field. As highlighted by Goude (2012), the flow curvature model would introduce a net positive angle of attack to the blade in the upwind half, which would redistribute the loads further upwind (Huang et al., 2023) and directly influence the wake.

**Reviewer 3**

*1. Choosing not to include the flow-curvature (virtual camber) effect to "avoid complicating error attribution" is questionable—excluding a known aerodynamic influence makes it harder, not easier, to understand discrepancies. A practical alternative would be to preprocess the airfoil polars by incorporating the effective camber caused by flow curvature under typical VAWT conditions. These adjusted polars could be used as CACTUS input, offering insight into the impact of virtual camber without modifying the code itself.*

Firstly, thank you for your critical feedback on the manuscript. We agree that not using the flowcurvature is a limitation in our work. We considered the possibility of preprocessing the airfoil polars using two methods: (1) using the equation from Goude (2012) where the polars are a function of the relative velocity $V_{rel}$, and (2) using the geometrical virtual camber algorithms or conformal mapping algorithms.

For method (1), if we preprocess the polars with a prescribed $V_{rel}$ distribution, we would encounter further errors as the wake is simulated by the free-wake vortex model. We pointed this out in our previous response to the referee comments, and in the Methodology section.

As for method (2), it is indeed possible to do this. But, as pointed out by van der Horst et al. (2016) (Figure 3), for high chord-to-radius ratios ($c/R > 0.2$), the existing virtual camber and conformal transformation models provide very different results. As most of the X-Rotor blades operate mostly at very high $c/R$ values, we believe it would induce more errors that have not been investigated before, and therefore leads to further uncertainty on the choice of models. Furthermore, as the angles of attack would be very high near the root sections, the errors would likely compound with the uncertainty of the polars at these high angles of attack. That is why we highlighted that these flow curvature models should be tested for the X-Rotor geometry before using them in high-quality scientific work. Once tested, the blade force profiles can be mapped to high-fidelity CFD results to gain insight into which approaches fit best. However, as agreed by Reviewer #2, this is currently out of the scope of the present study. Therefore, we have elaborated more on our justification of not using the flow curvature model for this study in the manuscript.

[Figure]

Figure 3: Virtual camber as a function of chord to radius ratio $c/R$ reproduced from van der Horst et al. (2016).

While this provides a more accurate depiction of airfoil motion in VAWTs, we opt not to include it here for two critical reasons. Primarily, implementing the flow-curvature model (such as Goude (2012)) to a rotor geometry with a large spanwise relative velocity distribution would make it exceedingly difficult to isolate the differences observed between CACTUS and experimental results to other factors. Moreover, geometrical virtual airfoil transformations (such as Hirsch and Mandal (1984)) show large inaccuracies at high chordto-radius ratios ($c/R > 0.2$), which is the regime that most of the X-Rotor blade operates in. Secondly, as CACTUS does not come inherently with a flow curvature correction model, implementing it ourself in the source code would be outside the scope of this study. Furthermore, attempting to pre-emptively correct the airfoil polars without calculating the relative velocities would result in unwarranted errors in CACTUS.

*2. The authors attribute wake discrepancies—particularly at X/D = 0.43—to the vortex core size, supported by a reference to an actuator line method (ALM) study. While there is a conceptual similarity between force smearing in ALM and vortex core effects in lifting line theory, the two methods are fundamentally different, and this comparison is not directly applicable. Concluding that the discrepancies stem from vortex convection assumes the aerodynamic loads are accurate—something not demonstrated in the paper. The brief mention of uncertainties due to airfoil polars is more relevant and deserves further exploration before focusing solely on the vortex core.*

The reviewer is indeed correct. In the original of this manuscript, we attributed the cause to the vortex core size as we ran into stability issues. Given that, in the second and the current version, we have demonstrated that the results are mostly independent to the vortex core size (refer sensitivity study in the Appendix), we should have updated our reasoning for these discrepancies. As the reviewer pointed out, most of them are mostly due to the large uncertainty in the airfoil polars. We have now updated all instances of the reasoning behind the discrepancies with further details on how they are affected by the airfoil polars. You can find the updated text directly referenced by this comment below. Furthermore, we have now elaborated more in Section 3.2 with plots to showcase the uncertainty in our choice of polars as well as the polars documented in Melani et al. (2019) at Re = 80K. Please refer to the marked-up document to fully track all the changes made.

At $X/D = 0.43$, CACTUS slightly under-predicts the velocity deficit across all azimuths. This difference is likely due to the challenge of uncertainty with the polars, as previously discussed in Section 3.2. If the polars were closer to the true experimental airfoil behaviour, the difference could be minimised. This holds true for the rest of the results discussed in this study.

*3. In the pitched case, where discrepancies grow more significant, the authors again attribute the issue to the vortex model while overlooking the potential role of load misrepresentation, such as inaccurate polars. This is a missed opportunity for a more balanced analysis.*
We agree with the reviewer. This has been addressed together with the previous comment throughout the manuscript.
These discrepancies are primarily attributed to differences in vortex strength predicted by CACTUS, which depend on the airfoil polars selected for the simulation. CACTUS struggles to accurately capture the increase in circulation induced by pitch offsets, as its predictions rely heavily on the input polars. Given that the static stall angle for these polars is $\alpha_{ss} = 9.1°$, pitching the airfoils by $\beta = 10°$ effectively shifts the $C_l$ vs $\alpha$ profile such that the blade would mostly operate in deep-stall conditions throughout its azimuth. As the accuracy of XFOIL becomes questionable at deep-stall conditions (Section 3.2), the errors are compounded.

*4. Finally, the sentence in the abstract—"Results indicate that CACTUS effectively replicates the flowfield within the rotor volume and the very near wake when no pitch offsets are applied"—overstates the level of agreement. While results align with experiments in some areas, noticeable discrepancies remain. A more measured phrasing would better reflect the findings.*
Thank you for pointing this out. We do agree with the reviewer that the sentence overstates the level of agreement. Therefore, we have now modified the sentence to be measured.
Results indicate that CACTUS is able to predict the flowfield to a reasonable extent within the rotor volume and in the very near wake when no pitch offsets are applied, with discrepancies attributed to the uncertainty of the polars at the low Reynolds numbers.

**References**

Goude, A.: Fluid Mechanics of Vertical Axis Turbines - Simulations and Model Development, ISBN 9789155485399, 2012.

Hirsch, C. and Mandal, A.: Flow Curvature Effect on Vertical Axis Darrieus Wind Turbine Having High Chord-Radius Ratio, in: European Wind Energy Conference, pp. 405–410, European Wind Energy Conference, 1984.

Huang, M., Sciacchitano, A., and Ferreira, C.: On the wake deflection of vertical axis wind turbines by pitched blades, Wind Energy, 2023.

Melani, P. F., Balduzzi, F., Ferrara, G., and Bianchini, A.: An annotated database of low Reynolds aerodynamic coefficients for the NACA0021 airfoil, in: AIP Conference Proceedings, vol. 2191, p. 20111, ISBN 9780735419384, ISSN 15517616, https://doi.org/10.1063/1.5138844, 2019.

van der Horst, S., van de Wiel, J. E., Ferreira, C. S., and García, N. R.: Flow Curvature Effects for VAWT: a Review of Virtual Airfoil Transformations and Implementation in XFOIL, in: 34th Wind Energy Symposium, American Institute of Aeronautics and Astronautics, ISBN 978-1-62410-395-7, https://doi.org/10.2514/6.2016-1734, 2016.